# Mapping the Topographic Features of Mining-Related Valley Fills Using Mask R-CNN Deep Learning and Digital Elevation Data

**Aaron E. Maxwell [1,*]**, **Pariya Pourmohammadi [2]** and **Joey D. Poyner [1]**

1   Department of Geology and Geography, West Virginia University, Morgantown, WV 26506, USA; jpoyner@mix.wvu.edu
2   Davis College of Agriculture, Natural Resources, and Design, Department of Design and Community Development, West Virginia University, Morgantown, WV 26506, USA; papourmohammadi@mix.wvu.edu
*   Correspondence: Aaron.Maxwell@mail.wvu.edu; Tel.: +1-304-293-2026

**Abstract:** Modern elevation-determining remote sensing technologies such as light-detection and ranging (LiDAR) produce a wealth of topographic information that is increasingly being used in a wide range of disciplines, including archaeology and geomorphology. However, automated methods for mapping topographic features have remained a significant challenge. Deep learning (DL) mask regional-convolutional neural networks (Mask R-CNN), which provides context-based instance mapping, offers the potential to overcome many of the difficulties of previous approaches to topographic mapping. We therefore explore the application of Mask R-CNN to extract valley fill faces (VFFs), which are a product of mountaintop removal (MTR) coal mining in the Appalachian region of the eastern United States. LiDAR-derived slopeshades are provided as the only predictor variable in the model. Model generalization is evaluated by mapping multiple study sites outside the training data region. A range of assessment methods, including precision, recall, and F1 score, all based on VFF counts, as well as area- and a fuzzy area-based user's and producer's accuracy, indicate that the model was successful in mapping VFFs in new geographic regions, using elevation data derived from different LiDAR sensors. Precision, recall, and F1-score values were above 0.85 using VFF counts while user's and producer's accuracy were above 0.75 and 0.85 when using the area- and fuzzy area-based methods, respectively, when averaged across all study areas characterized with LiDAR data. Due to the limited availability of LiDAR data until relatively recently, we also assessed how well the model generalizes to terrain data created using photogrammetric methods that characterize past terrain conditions. Unfortunately, the model was not sufficiently general to allow successful mapping of VFFs using photogrammetrically-derived slopeshades, as all assessment metrics were lower than 0.60; however, this may partially be attributed to the quality of the photogrammetric data. The overall results suggest that the combination of Mask R-CNN and LiDAR has great potential for mapping anthropogenic and natural landscape features. To realize this vision, however, research on the mapping of other topographic features is needed, as well as the development of large topographic training datasets including a variety of features for calibrating and testing new methods.

**Keywords:** light detection and ranging; LiDAR; deep learning; convolutional neural networks; CNNs; mask regional-convolutional neural networks; mask R-CNN; digital terrain analysis; resource extraction

## 1. Introduction

Light detection and ranging (LiDAR) data provide high spatial resolution, detailed representations of bare earth landscapes, and have been shown to be valuable for mapping features of geomorphic

and archeological interest. For example, Joboyedoff et al. [1] suggest that LiDAR is an essential tool for detecting, characterizing, monitoring, and modelling landslides and other forms of mass movement. Chase et al. [2] argue that LiDAR technologies have resulted in a paradigm shift in archeological research, as they allow for the mapping of ancient anthropogenic features and landscapes even under dense canopy cover. For example, LiDAR has recently improved our understanding of ancient Mesoamerican cultures by mapping ancient cities now obscured by dense forest cover, a mapping task that is too labor intensive for field-based survey methods alone [2]. Further, LiDAR data are becoming increasingly available for public download, especially in Europe and North America. For example, the United States has implemented the 3D Elevation Program (3DEP) (https://www.usgs.gov/core-science-systems/ngp/3dep) with a goal of providing LiDAR coverage for the entire country, excluding Alaska [3]. In this spirit, The Earth Archive project has argued for the need for 3D data of the entire Earth surface to create a historic record for future generations, and is currently soliciting donations to support this project [4].

Despite the increasing availability of high spatial resolution digital terrain data, and the wealth of information that can be derived from such data, the extraction of features from these data to support archeological, geomorphic, and landscape change research is in many cases dominated by manual interpretation, as previously noted by [5,6]. With the exception of some notable studies (e.g., [6–9]), generic and automated mapping of topographic features from digital elevation data has proved to be a particularly challenging task. However, deep learning (DL), and in particular mask regional-convolutional neural networks (Mask R-CNN), may make it possible to realize the potential of digital elevation data for automated mapping of topographic features.

Therefore, this study explores the use of Mask R-CNN for mapping valley fill faces (VFFs) resulting from mountaintop removal (MTR) surface coal mining in the Appalachian region of the eastern United States. MTR is a common mining method in this region which results in extensive modifications to the landscape, and therefore mapping VFFs is of significant interest for environmental modelers. From our findings, we comment on the application of this DL method for extracting anthropogenic and natural terrain features from LiDAR-derived data based on distinct topographic and spatial patterns. Since LiDAR data are not commonly available to represent historic terrain conditions due to only recent development of this technology for mapping large spatial extents, we also explore the transferability of the models to older, photogrammetrically-derived elevation data. This study therefore has two objectives:

1. Assess the Mask R-CNN DL algorithm for mapping VFFs using LiDAR-derived digital elevation data.
2. Investigate model performance and generalization by applying the model to LiDAR-derived data in new geographic regions and acquired with differing LiDAR sensors and acquisition parameters, as well as a photogrammetrically-derived digital terrain dataset.

### 1.1. LiDAR and Digital Terrain Mapping

LiDAR is an active remote sensing method that relies on laser range finding. A laser pulse is emitted by a sensor. When the emitted photons strike an object, a portion of the energy is reflected back to the sensor. Using the two-way travel time of reflected laser pulses detected by the sensor, global position system (GPS) locations, and aircraft orientation and motion from an inertial measurement unit (IMU), horizontal and elevation coordinates of the reflecting surface can be estimated at a high spatial resolution. Further, a single laser pulse can potentially result in multiple returns, allowing for vegetation canopy penetration and the mapping of subcanopy terrain, in contrast to other elevation mapping methods [10].

LiDAR data have been applied to a variety of terrain mapping and analysis tasks. For example, many studies have investigated the mapping of slope failures, such as landslides, using terrain variables derived from LiDAR [1,8,11–13]. Another common application is modeling the likelihood of slope failure occurrence or landslide risk [14–17]. In a 2012 review of the use of LiDAR in landslide

investigations, Jaboyedoff et al. [1] suggest that LiDAR is an essential tool for landslide risk management and that there is a need to develop methods to extract useful information from such data. Older, photogrammetrically-derived elevation data have also been used for terrain mapping and analysis tasks and offer a means to characterize historic terrain conditions. For surface mine mapping specifically, Maxwell and Warner [18] found that historic, photogrammetric elevation data were of great value for differentiating grasslands resulting from mine reclamation from other grasslands while DeWitt et al. [19] provided a comparison of different digital elevation data sources for mapping terrain change resulting from surface mining.

Object-based image analysis (OBIA) has been applied to LiDAR data for the mapping of landslides [20] and geomorphic landforms in general [9]. OBIA incorporates segmentation of raster-based data into regions or polygons, based on measures of similarity or homogeneity. These polygons are the spatial unit of analysis and classification [21]. Part of the interest in OBIA for geomorphic mapping is the ability to incorporate spatial context information into the mapping process, facilitated by the data segmentation. Nevertheless, choosing the scale of the segmentation is a major hurdle in OBIA, and indeed some research indicates it is necessary to choose multiple scales [22,23]. In contrast, spatial context information can be included in DL by using convolutional neural networks (CNNs) in a manner that does not require a priori specification of the scale. Thus, applying CNN-based DL to digital terrain data holds great promise.

## 1.2. Deep Learning

DL algorithms are derived from, and offer an extension to, artificial neural networks (ANNs). Traditional ANNs generally have a small number of hidden layers, whereas DL algorithms have many hidden layers. In contrast to traditional machine learning methods, which are shallow learners, it has been suggested that DL is able to provide a higher level of data abstraction, potentially resulting in improved predictive power, generalization, and transferability [24–30]. Although this results in a model that is much more complex and has many more parameters, it allows for multiple levels of data abstraction to learn complex patterns. Like other supervised machine learning methods, DL requires example training data with associated labels in order to build the model. A measure of error or performance, generally termed loss, is used to guide the algorithm to improve predictions as it iterates through the training data multiple times [24,30].

CNNs extend the deep ANN architecture to incorporate context information into the prediction. CNNs include convolutional layers that learn filters that transform input image values, similar to moving window kernels traditionally used in remote sensing for image edge detection and smoothing. However, in the case of CNNs, the algorithm produces optimal filters to aid in predicting the labels associated with the training images. The addition of this context information has offered substantial advancements in computer vision and scene labeling problems [24,28,30]. In remote sensing applications, CNNs allow for the analysis of spatial context information when applied to high spatial resolution data (for example, [28]), spectral patterns when applied to hyperspectral data (for example, [31]), and temporal patterns when applied to time series products (for example, [32]). Thus, DL with convolution allows for the integration of contextual information in the spatial, spectral, and temporal domains.

Traditional CNNs have primarily been used for scene labeling problems, for example, entire images or image chips categorized by different land cover type. Traditional CNNs do not allow for pixel-level or semantic labeling. However, the introduction of fully convolutional neural networks (FCNs) alleviated this limitation by combining convolution and deconvolution layers with up-sampling, which allows for the final feature map to be produced at the original image resolution with a prediction at each cell location [27,33], similar to traditional remote sensing classification products. Example FCN architectures include SegNet [34] and UNet [35–37].

In this study, we use instance segmentation methods, in which the goal is to distinguish each individual instance of a feature in the scene separately. For example, each tree in a scene can be

identified as a separate instance of the tree class. We specifically implement the Mask R-CNN method. This method is an extension of faster R-CNN, which allows for convolution to be applied on regions of the image as opposed to the entire scene. This involves generating convolution feature maps that are then applied to individual subsets of the image, called regions of interest (RoI), defined by the region proposal network (RPN). The process of RoI pooling allows convolution features to be applied to regions of the image of different sizes and rectangular shapes [38,39]. Mask R-CNN extends this framework to allow for polygon masks to be generated within each RoI, essentially performing semantic segmentation within each RoI using FCNs. This requires better alignment between the RoI pooling layers and the RoIs than is provided by faster R-CNN. So, a ROIAlign layer is applied to improve the spatial alignment [39]. Since there are multiple components of the model, multiple loss measures are used to assess performance. Specifically, the total loss is the sum of the loss for the bounding box, classification, and mask predictions [38,39]. For a full discussion of Mask R-CNN, please consult He et al. [39], who introduced this method.

DL methods have shown promise in remote sensing mapping and data processing tasks including scene labeling, pixel-level classification, object-detection, data fusion, and image registration [24]. For example, Microsoft has recently used DL to map 125 million building footprints across the entire US [40]. Kussul et al. [26] explored DL for differentiating crops using a time series of Landsat-8 multispectral and Sentinel-1 synthetic-aperture radar (SAR) data and documented improved overall and class-specific classification performance in comparison to shallow learners, such as random forests (RF). Li et al. [41] used DL and QuickBird satellite imagery to map individual oil palm trees with precision and recall rates greater than 94%.

It should be noted that there are some complexities in implementing these methods and applying them to remotely sensed data, such as the need for a large number of training samples, the difficulty of model optimization and parameterization, and large computational demands [24,30]. Also, the processes of training models and predicting to new data can differ from those used in traditional image classification and machine learning; for example, convolution requires training on and predicting to small rectangular image extents, or image chips, as opposed to individual pixels or image objects. Thus, researchers and analysts must augment workflows and learn new techniques for implementing DL algorithms [24,30].

A review of the literature suggests that the application of DL to LiDAR and digital terrain data is still limited. There has been some research relating to using DL for extracting ground returns from LiDAR point clouds for digital terrain model (DTM) generation (for example, [33,42–44]). Specifically, Hu and Yuan [44] suggest that DL-based algorithms can outperform the current methods that are most commonly used for ground return classification. Others have investigated the classification of features in 3D space represented as point clouds [45–47].

There is a need to investigate mapping anthropogenic and natural terrain features from digital terrain data using DL, as the research on this topic is currently lacking; however, there have been some notable studies. Tier et al. [6] investigated the identification of prehistoric structures from LiDAR-derived raster data. From the LiDAR data, a DTM was interpolated followed by a measure of local relief, which was then provided as input to the ResNet18 CNN algorithm as an RGB image. They reported mixed results, with some areas predicted well and other areas suffering from many false positives. Behrens et al. [48] explored digital soil mapping using DTM raster data and DL and obtained a more accurate output than that produced by RF.

Interestingly, a number of studies attempt to map features that are at least partially characterized by geomorphic and terrain characteristics using spectral data only, without using terrain data. For example, Li et al. [49] mapped craters from image data using faster R-CNN and obtained a mean average precision (mAP) higher than 0.90. As an example a study that combined spectral and terrain data, Ghorbanzadeh et al. [50] used RapidEye satellite data and measures of plan curvature, topographic slope, and topographic aspect to detect landslides. They noted comparable performance between CNNs and traditional shallow classifiers: ANN, SVM, and RF.

Mask R-CNN has seen limited application in remote sensing at the time of this writing. Zhang et al. [51] assessed the method for mapping artic ice-wedge polygons from high spatial resolution aerial imagery and documented that 95% of individual ice-wedge polygons were correctly delineated and classified, with an overall accuracy of 79%. Zhao et al. [37] found that Mask R-CNN outperformed UNet for pomegranate tree canopy segmentation. Stewart et al. [52] used the method to detect lesions on maize plants from northern leaf blight using unmanned aerial vehicle (UAV) data. Given the small number of studies that have applied this algorithm to remotely sensed data, there is a need for further exploration of this algorithm within the discipline. We found a lack of research associated with mapping terrain features from digital terrain data using DL methods, and no published studies that apply this algorithm to raster-based digital terrain data for mapping geomorphic features. We attribute this to the only recent advancement of DL for semantic and instance segmentation, and lack of available data to train DL models.

### 1.3. Mountaintop Removal Coal Mining and Valley Fills

In this study we apply Mask R-CNN to detect instances of VFFs from digital terrain data derived from LiDAR. Valley fills are a product of MTR coal mining, which has been practiced in southern West Virginia, eastern Kentucky, and southwestern Virginia in the Appalachian region of the eastern United States for several decades. This surface mining process involves using heavy machinery to extract thin interbedded coal seams. Valley fills are generated from the redistribution of overburden rock material. Since the coal seams are interbedded with other rock types of limited commercial value, a large volume of displaced material is produced. Due to the original steepness of the slopes, it is not possible to reclaim the landscape to the approximate original contour. Therefore, excess overburden material is placed in adjacent valleys, raising the valley elevation and changing the landscape.

The excavation and subsequent reclamation associated with valley fills results in substantial alterations to land cover, soil, and the topography and contour of the landscape [53–62]. Forests are lost and fragmented [63], mountaintop elevations are lowered by tens to hundreds of meters [53,57]; soils are compacted [58,64], and human quality of life and health is affected by exposure to chemicals, dust, and particulates [62]. Because valley fills bury headwater streams [54,61], and the fill material is hydrologically dissimilar to undisturbed land, hydrology is particularly affected. Valley fills tend to increase stream conductivity and alter hydrologic regimes downstream [53,56,58,64]. Wood and Williams [61] documented a decrease in salamander abundance in headwater streams impacted by valley fills in comparison to reference streams. In summary, valley fills profoundly alter the landscape, resulting in a variety of complex effects on the physical environment and its inhabitants, making it of vital importance that these features be monitored and mapped over time to facilitate environmental modeling.

Figure 1 provides examples of valley fills within the study area. Note that these features are generally characterized by steep slopes, a terraced pattern to encourage stability, placement in headwater stream valleys adjacent to mines and reclaimed mines, and drainage ditches to transport water away from the mine site. In short, they have a unique topographic signature and are readily observable in digital terrain data representations, such as hillshades and slopeshades. Due to this unique signature and their potential environmental impacts, we argue that this is a valuable case study in which to assess the use of Mask R-CNN for detecting and mapping topographic features. Here, we are specifically attempting to map the valley fill faces (VFF; i.e., the graded slope that faces the downstream valley not yet filled). Since the true extent of the filled area and excavated areas are not readily observable and grade into one another, the upper extent of each fill is hard to distinguish. Therefore, we focus exclusively on the VFF.

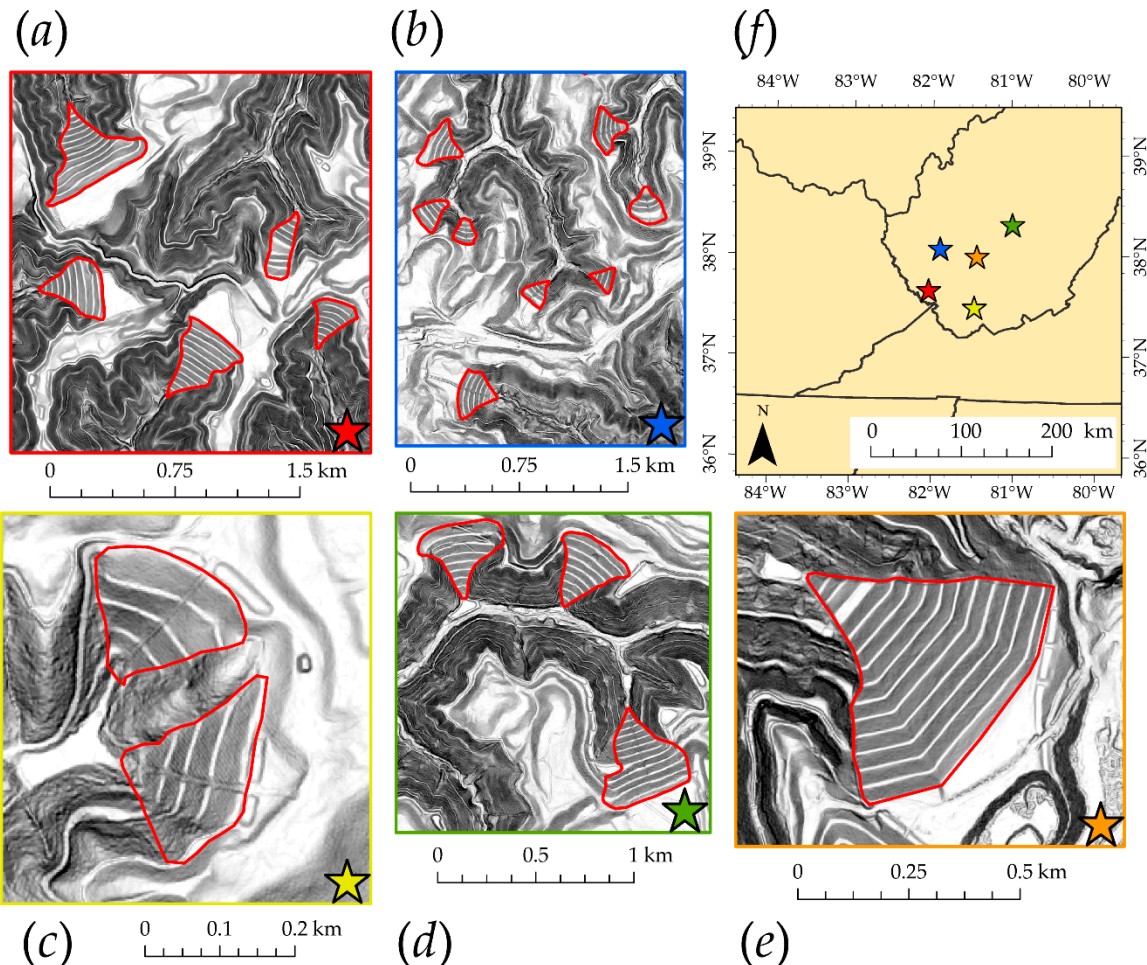

**Figure 1.** Example of valley fill faces (VFFs) within study area extent. The images are slopeshades, generated from the light-detection and ranging (LiDAR) data. Stars indicate the location of (*a*) through (*e*) in the study areas (*f*).

## 2. Methods

### 2.1. Study Area and Training Data Digitizing

The study areas are shown in Figure 2. The training, testing, and validation data partitions are nonoverlapping and defined geographically. The training area (Train) has a size of 9019.6 km² and occurs completely within West Virginia. Areas adjacent to the training area, mapped with the same LiDAR sensor and also within West Virginia, were withheld to test model performance during the training process (Test) and to validate the final model once obtained (Val). In order to assess how well the model generalizes to new LiDAR-derived terrain data, we performed additional validations over two areas in Kentucky (KY1 and KY2) and one area in Virginia (VA). Digital terrain data produced using photogrammetric methods were also predicted including a subset of the training area (SAMB1) and the entire validation area (SAMB2); no photogrammetric data were used in the training dataset. In summary, the model was trained over a single large area, tested over an adjacent smaller area, then used to make predictions in a new area collected with the same LiDAR sensor, three additional extents in different states mapped during different LiDAR collections, and two extents of photogrammetric data within West Virginia.

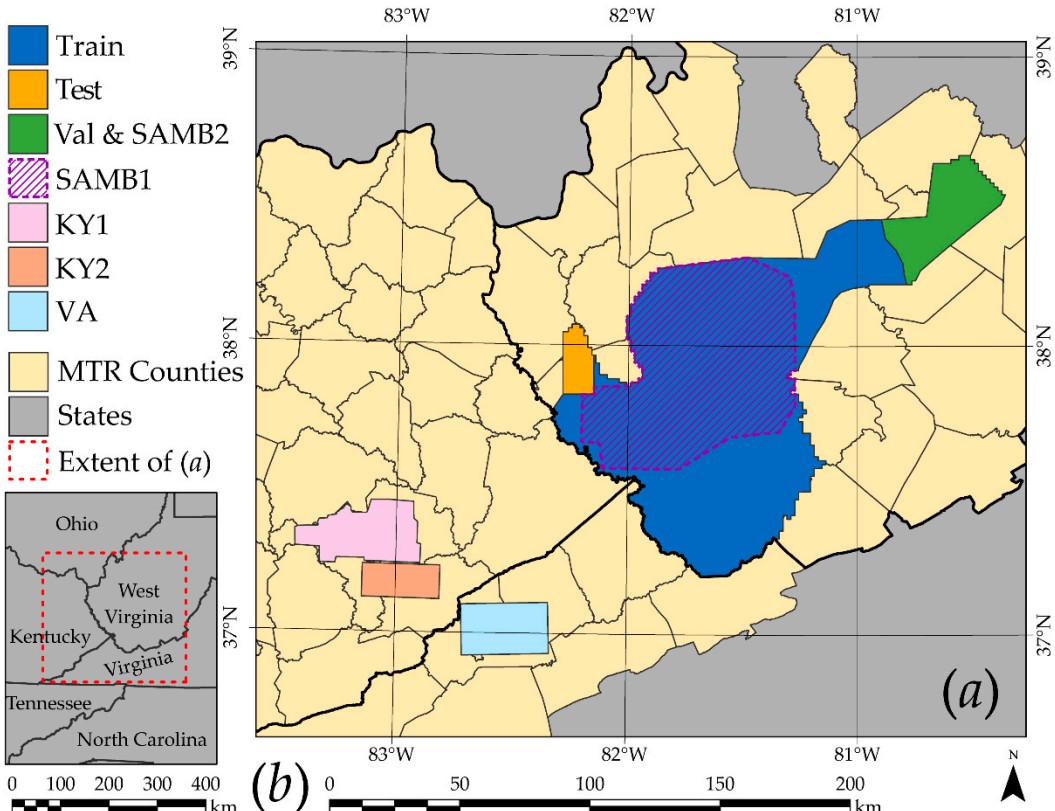

**Figure 2.** (*a*) Study areas in West Virginia, Kentucky, and Virginia in the Appalachian region of the eastern United States. (*b*) shows the extent of (*a*) in the eastern United States.

The extents were primarily chosen based on the availability of LiDAR data and the abundance of VFFs. In total, 1105 VFFs were provided to train the model, 118 were used to test the model at the end of each epoch in the training process, and 1014 were used to validate the model over different geographic extents (Table 1). Training data were digitized by two analysts by visual interpretation of LiDAR-derived terrain surfaces, specifically hillshades and slopeshades, and additional geospatial data, including aerial imagery. Based on the size distribution of digitized VFFs, a minimal mapping unit (MMU) of 0.2 ha was defined for this study and no VFFs smaller than this size were included.

**Table 1.** Description of study areas and mapped VFFs.

| Study Area | Total Area | Number of Image Chips with Valley Fills | Number of Image Chips to Predict To | Number of Valley Fills |
|---|---|---|---|---|
| Train | 9019.6 km$^2$ | 4863 | - | 1105 |
| Test | 279.5 km$^2$ | 282 | - | 118 |
| Val | 921.0 km$^2$ | - | 3111 | 182 |
| KY1 | 773.4 km$^2$ | - | 2650 | 540 |
| KY2 | 338.9 km$^2$ | - | 1138 | 149 |
| VA | 599.3 km$^2$ | - | 2093 | 143 |
| SAMB1 | 4661.8 km$^2$ | - | 17,106 | 581 |
| SAMB2 | 921.0 km$^2$ | - | 3110 | 108 |

*2.2. Input Terrain Data and Pre-Processing*

Descriptions of the LiDAR data and collections are provided in Table 2. The West Virginia LiDAR data used in this study were obtained as classified point clouds from the West Virginia View/West Virginia GIS Technical Center Elevation and LiDAR Download Tool (http://data.wvgis.wvu.edu/elevation/). LiDAR data for the study sites in Kentucky and Virginia were downloaded from the

3DEP website (https://www.usgs.gov/core-science-systems/ngp/3dep) also as classified point clouds. Data and collection information were obtained from the associated metadata files. Although all data were collected during leaf-off conditions, they differ based on collection dates, sensor used, sensor specifications, and flight specifications. The West Virginia and Kentucky data provide similar point densities at an average of 1 point per square meter (ppsm) while the Virginia data offer a higher point density at 1.7 ppsm. Collectively, the data were collected over nearly seven years. In summary, there are many differences in these datasets to support our goal of assessing transferability of models to new data and geographic extents.

**Table 2.** Description of LiDAR data.

| | LiDAR Dataset | | |
|---|---|---|---|
| **Specification** | **West Virginia** | **Kentucky** | **Virginia** |
| Collection Dates | 4-9-2010 to 12-31-2011 | 11-8-2011 to 1-19-2013 | 11-3-2016 to 4-17-2017 |
| Phenology | Leaf-off | Leaf-off | Leaf-off |
| Sensor | Optech ALTM-3100 | Leica ALS70 and Optech Gemini | Riegel 780/680i |
| Average Post Spacing | 1 ppsm | 1 ppsm | 1.746 ppsm |
| Flight Height | 1524 m AGL | 1828 m AGL | 1800 m AGL |
| Approximate Flight Speed | 135 knots | 116 knots | 100 knots |
| Scanner Pulse Rate | 70 kHz | 50 kHz | 280 kHz |
| Scan Frequency | 35 Hz | 30.1 Hz | 68 Hz |
| Maximum Scan Angle | 36° | 25.6° | 60° |

AGL = above ground level, ppsm = points per square meter.

The 0.61 m (2 ft) true color stereo imagery used to derive the photogrammetric elevation data used in this study were collected during the spring of 2003 and 2004 during leaf-off conditions as part of a mapping project supported by the West Virginia Statewide Addressing and Mapping Board (WVSAMB). Break lines and elevation mass points were generated using photogrammetric methods at a 3 m interval with a vertical accuracy of ±10 ft. The final 3 m DEM has a tested vertical accuracy of 0.209 m [65].

All LiDAR point clouds were converted to raster grids as DTMs using only the points classified as ground returns and the LAS Dataset to Raster utility in ArcGIS Pro 2 [66]. The average ground return elevation was calculated within each cell, and linear interpolation was used to fill data gaps. Based on data volume and visual assessment of outputs, we gridded all data to a 2 m resolution, as the VFFs and their topographic signature were easily discernable at this spatial resolution. The photogrammetric data were resampled from 3 m to 2 m using cubic convolution to match the LiDAR data.

Many surfaces can be derived from DEMs to characterize and visualize the terrain [67]. We experimented with multiple terrain visualization methods including hillshades, multi-directional hillshades, hypsometrically-tinted hillshades, and slopeshades. Based on visual inspection and initial experimentation with the Mask R-CNN algorithm, the slopeshade was selected to represent the terrain surface because it provides a distinctive and relatively consistent representation of VFFs, unlike hillshades, which vary in appearance based on the local angle of illumination of the solar energy. Since data augmentation, including random rotations and flips of the data, are used to minimize overfitting in this study, as will be discussed below, an illumination invariant terrain representation is preferred.

Slopeshades are generated from a topographic slope surface. A grayscale color ramp is applied to represent steep slopes with darker shades and flat areas with brighter shades [15,68–70]. To produce these surfaces, we first calculated topographic slope in degrees using the Slope Tool from the Spatial

Analyst Extension of ArcGIS Pro 2 [66]. The data were then converted to 8-bit integer data using Equation (1). We used a maximum slope of 90° as opposed to the maximum value in each grid surface so that all study sites could be rescaled consistently.

$$\text{Slopeshade} = (1 - \frac{\text{Slope}}{90}) \times 255 \tag{1}$$

### 2.3. Image Chip Generation

Since spatial context information is learned using filters, DL methods that include convolution must be trained on rectangular image chips as opposed to single pixels or image objects [24,28]. Image chips were generated using the Export Training Data For Deep Learning Tool in ArcGIS Pro 2 [66]. In order to provide training and testing data for the Mask R-CNN algorithm, the geographic extents were segmented into 512-by-512 pixel image chips. We applied a stride of 256 pixels in the X- and Y-directions for overlap and to produce more training and testing data. Using this method, 4863 training and 282 testing chips were generated that contained at least one instance of the VFF class, as noted in Table 1 above. Training masks were also generated for each image chip using this tool. Background or non-VFF pixels were coded to 0, while each instance of VFFs was coded with a unique value, from 1 to the number of VFFs in the extent, as demonstrated in Figure 3. Instance segmentation methods, such as Mask R-CNN, require that each unique instance be differentiated whereas semantic methods can accept a binary mask to differentiate a class from background pixels [34,35,39]. Once a final model was obtained, it was used to predict VFFs from image chips covering the Val extent and all other study areas. Initial results showed poor predictions near the edge of image chips. To circumvent this issue, we made predictions using larger image chips, with dimensions of 1024-by-1024 pixels and with a stride of 256 pixels, allowing for substantial overlap so that features were not missed and were likely to occur near the center of at least one chip. We then cropped each chip so that only the center 50% was used in the final surface.

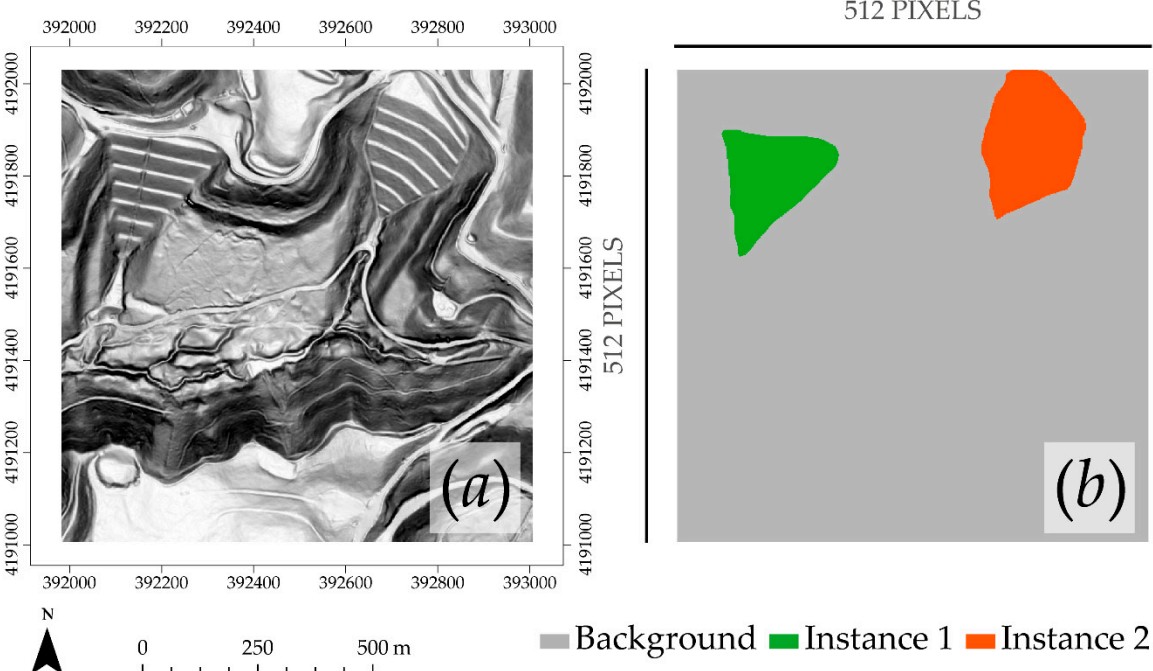

**Figure 3.** Example of image chip (*a*) and associated training mask with two mapped VFFs (*b*). The coordinate grid in (*a*) represents easting and northing relative to the NAD 83 UTM Zone 17 North projection.

*2.4. Mask R-CNN Implementation*

We used the Matterport implementation of Mask R-CNN in this study, which is available on GitHub (https://github.com/matterport/Mask_RCNN). This implementation uses Python 3, Keras, and Tensorflow. In order to load our image chips and masks, we had to generate a subclass of the Dataset class provided by the implementation. All experiments were conducted on a workstation equipped with an i9 7900X 3.3 GHz 10 core processor, 128 GB of RAM, and a GeForce RTX 2080 Ti 11 GB graphics card.

To train the model, we used a learning rate of 0.002 to train the head layers for 2 epochs, followed by training all layers at a learning rate of 0.001 for 12 epochs, and lastly training all layers at a learning are of 0.0001 for an additional 10 epochs. An initial experiment was conducted in which learning progressed for 85 epochs. In this experiment, we observed overfitting early and found that 24 epochs were adequate to stabilize the model. The default learning momentum and weight decay values, 0.9 and 0.0001, were used for all epochs. We also maintained the default values for the backbone strides of the feature pyramid network (FPN), RPN anchor scales, RPN anchor scale ratios, and RoI positive ratio. Mask R-CNN includes ROIs for training that do not contain an example of the feature of interest so that it can also learn from negative cases. In this study an ROI positive ratio of 33% was used, or 67% represented negative cases. All loss measures were equally weighted. We use the ResNet101 backbone [71] to learn the convolution filters. For each epoch, 1500 training and 90 validation steps were used with a batch size of 3 image chips, allowing for 4500 training samples and 270 test samples to be used in each epoch. The model that produced the best loss value for predicting to the test samples was selected as the final model.

Initial experimentation with initializing the model from random weights was unsuccessful, perhaps because we did not provide enough training samples to adequately train the complex model [24,30,39]. As a result, we initialized the model using weights learned from the Microsoft Common Objects in Context (MS COCO) dataset [72] (http://cocodataset.org/#home), a process known as transfer learning. Many studies have noted the value of initializing models using pre-trained weights learned from other data and problems, even if the data and classes are different [6,24,30,38,51,73–75]. For example, Tier et al. [6] used pre-trained weights learned from photographs to initialize a model to extract archeological features form digital terrain data. For Mask R-CNN specifically, Zhang et al. [51] initialized their model from the COCO weights for mapping artic ice-wedge polygons from very high spatial resolution aerial imagery. The argument for applying transfer learning is that low-level data abstractions learned from imagery can be valuable when applied even to disparate data [24,30,38,39,74,75]. Since the MS COCO weights were obtained relative to RGB images, the slopeshade data were loaded in as 3-band images by replicating the grayscale values to each band. Although not computationally efficient, this allowed us to make use of transfer learning.

Data augmentation has been shown to minimize overfitting by expanding the number and characteristics of the training samples [24,29,30,50]. Therefore, we implemented random augmentations of the original image chips including rotations at 0°, 90°, 180°, and 270°; left/right flips; up/down flips, brightness and contrast alterations, and blurring and sharpening. We attempted to avoid extreme augmentations of the original data. These augmentations were applied using the imgaug Python library [76]. Example augmentations for the image chip shown in Figure 3 are provided as an example in Figure 4.

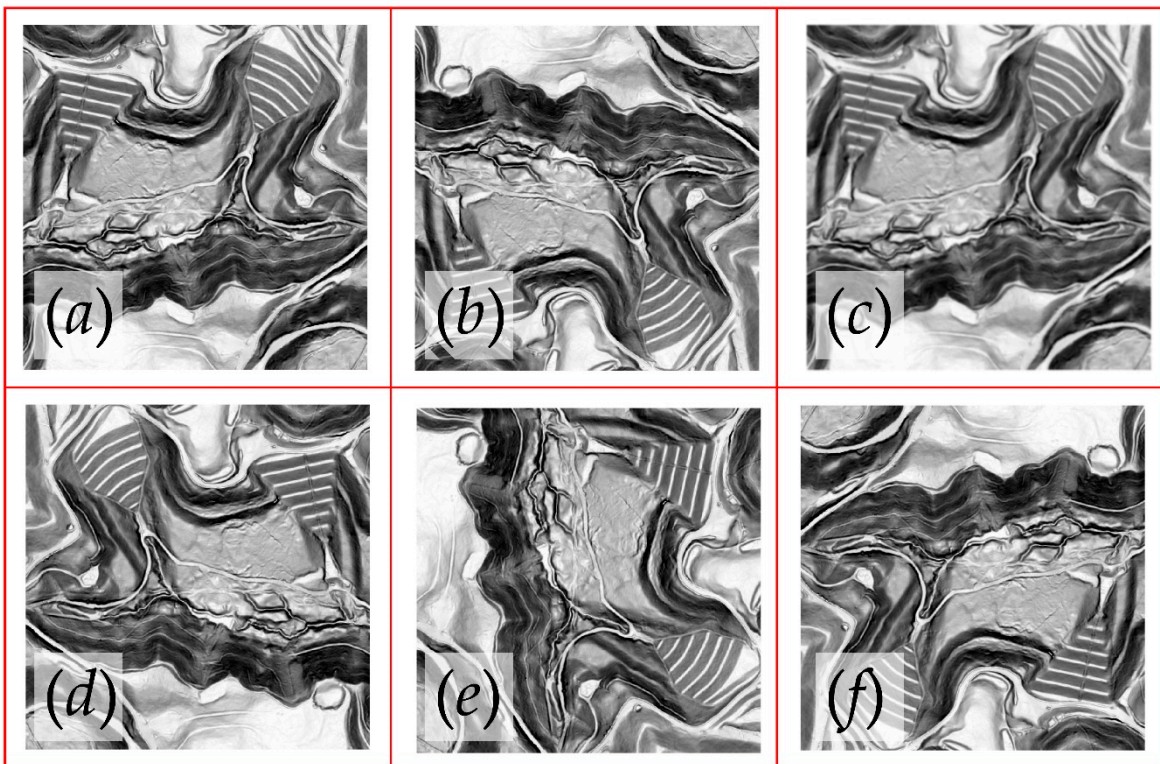

**Figure 4.** Example image augmentations applied to minimize overfitting. (*a*) Original image chip. (*b*) Vertical flip. (*c*) Gaussian blur. (*d*) Horizontal flip. (*e*) Rotate 90° clockwise. (*f*) Rotate 180° clockwise.

### 2.5. Prediction and Post-Processing

Once a final model was obtained, it was used to detect features in the Val extent and all other study areas. The final chips were binary surfaces, in which all predicted VFFs were coded to 1 and the background was coded to 0. This process was completed using the Matterport Mask R-CNN code combined with additional Python and R [77] scripts.

Once all image chips within a dataset were processed by the model and cropped, they were merged to a continuous raster surface using the Mosaic to New Raster utility in ArcGIS Pro [66] with the maximum value returned at cells in overlapping area so that all predicted VFF pixels that occurred in the center of the chips were maintained in the final model. The raster grids were then converted to polygon vectors to represent each contiguous area of VFFs as a single feature. Any predicted features smaller than 0.2 ha were removed to satisfy the MMU.

### 2.6. Accuarcy Assessment

We assessed the Matterport Mask R-CNN model based on mAP at multiple intersection of union (IoU) threshold ranges. IoU is the area of intersection divided by the area of union between the manually digitized and predicted masks as described in Equation (2). mAP represents the interpolated precision at multiple IoU threshold ranges based on the area under the precision-recall curve [78].

$$\text{IoU} = \frac{\text{Area of Intersection}}{\text{Area of Union}} \tag{2}$$

Given that the final output of the classification was vector objects occurring over a geographic extent, assessment using overlapping image chips, where the same VFF has the potential to be mapped and evaluated multiple times, will not yield an assessment that approximates the accuracy of the final map product. In this study, our primary interest is the detection of VFFs across the entire dataset as discrete spatial objects. Thus, we focus on true positives (TP), false positives (FP), and false negatives

(FN) [79] in the map. An additional complexity is that the boundaries of VFFs are inherently fuzzy at the fine scale of our map, which has 2 m pixels. Therefore, we assessed the predictions based on a visual comparison with the manually digitized VFFs. If a predicted feature was judged to overwhelming agree with the manually digitized VFF based on shared area and spatial co-occurrence, then it was labelled as a TP. FPs represent areas mapped as VFFs but were in reality not. FNs represent VFFs that were missed.

From the TP, FP, and FN counts of VFFs within each study area extent, we calculated precision, recall, and the F1 score. Precision represents the portion of the predicted VFFs that were VFFs and is equivalent to 1 - commission error. Recall represents the ratio of correctly mapped VFFs relative to the total number of VFFs and is equivalent to 1 - omission error. The F1 score is the harmonic mean of precision and recall. The equations for these metrics are provided below in Equations (3) through (5). We also assessed the TP, FN, and recall for VFFs that were larger than 1 ha in the manually digitized data to explore the performance for only larger features.

$$\text{Precision} = \frac{\text{TP}}{\text{TP} + \text{FP}} \tag{3}$$

$$\text{Recall} = \frac{\text{TP}}{\text{TP} + \text{FN}} \tag{4}$$

$$\text{F1 Score} = 2 \times \frac{\text{Recall} \times \text{Precision}}{\text{Recall} + \text{Precision}} \tag{5}$$

In addition to evaluating the numbers of correctly mapped VFFs, we also assessed the accuracy based on the delineation of VFFs, by focusing on the intersection and union of the derived polygons to estimate area-based producer's and user's accuracy. Using area based measures, larger VFFs have a larger weight in the assessment. Producer's accuracy is similar to recall, while user's accuracy is similar to precision [80–82].

Due to the indeterminate nature of VFF boundaries at the 2 m scale of this project, small differences between the location of the boundary in the reference and map data have little significance. Therefore, we also assessed producer's and user's accuracy for the VFF class using a fuzzy accuracy method modified from Brandtberg et al. [83]. The aim of this approach is to weight the disagreement between predicted and reference polygons, based on distance from the center of the polygon. Thus, a larger weighting is assigned to pixels near the center, and a lower weighting for pixels at the boundary. The fuzzy accuracy was implemented by first creating raster-based Euclidean distance surfaces at a 2 m spatial resolution using ArcGIS Pro [66] for each reference and predicted VFF polygon to represent the straight-line distance of each cell from the feature boundary. We divided each pixel in the VFF feature by the sum of the distances within the polygon, then multiplied the pixel by the area of the polygon, thus achieving our aim of weighting the entire area of the feature higher in the center in comparison to the edges, but keeping the data on a scale that is equivalent to area. We then summed the pixel values in the union and intersecting extents to obtain fuzzy estimates of producer's and user's accuracy for the VFF class from these totals.

## 3. Results

### 3.1. Mask R-CNN Model and Visual Assessment

Figure 5 shows loss measures for different components of the Mask R-CNN model as a function of epoch. The lowest overall loss calculated from the withheld test data was obtained after 12 epochs (0.773), so the result from this epoch was chosen for the final model. The graphs suggest some overfitting after 12 epochs; however, the dominant pattern is variability in the test loss measures and no substantial increase in performance. Other authors have noted optimal performance after few epochs, especially when pre-trained weights are used. For example, Zhang et al. [51] note optimal performance after eight epochs when using Mask R-CNN to predict artic ice wedge extents from high

spatial resolution aerial imagery, and Stewart al. [52] used only six epochs to map northern leaf blight lesions from UAV data.

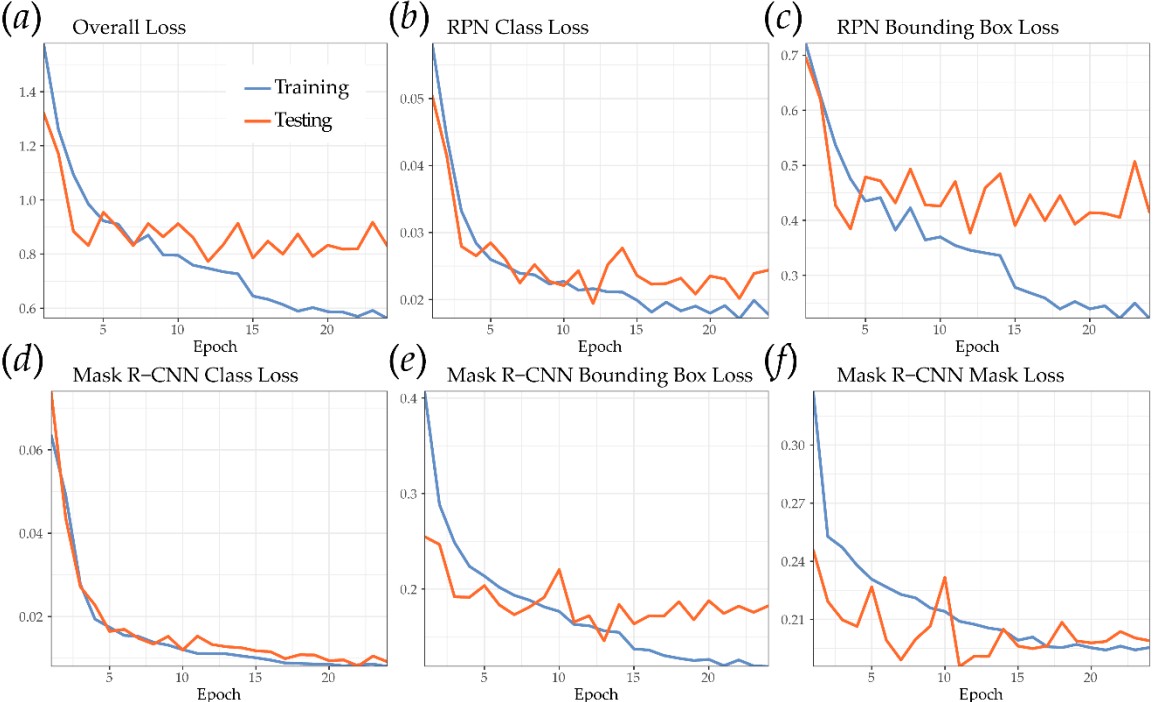

**Figure 5.** Loss values for training and test data across all epochs. (*b*) region proposal network (RPN) class loss measures how well the Region Proposal Network separates the background from the objects of interest. (*c*) RPN bounding box loss assesses how well the RPN localizes objects. (*d*) Mask R-CNN class loss assesses how well Mask R-CNN recognizes each class of object. (*e*) Mask R-CNN bounding box loss measures how well Mask R-CNN localizes objects. (*f*) Mask R-CNN mask loss measures how well Mask R-CNN segments objects. (*a*) Overall loss is an addition of all other loss measures. For a complete explanation of these metrics see [38,39].

Table 3 shows the mAP results for different IoU threshold ranges calculate from image chips covering the Val area. Performance decreased as the threshold was adjusted to incorporate higher IoU values, as expected. When only IoU thresholds between 0.50 and 0.55 were used, the mAP was 0.596. This generally suggests that with this narrow threshold range, VFFs were detected, but the boundaries did not match well, due to the indeterminate and complex nature of the VFFs.

**Table 3.** mAP results for validation data using different intersection of union (IoU) ranges.

| Start (IoU) | End (IoU) | mAP |
|:---:|:---:|:---:|
| 0.50 | 0.95 | 0.389 |
| 0.50 | 0.90 | 0.433 |
| 0.50 | 0.85 | 0.475 |
| 0.50 | 0.80 | 0.475 |
| 0.50 | 0.75 | 0.535 |
| 0.50 | 0.70 | 0.557 |
| 0.50 | 0.65 | 0.557 |
| 0.50 | 0.60 | 0.596 |
| 0.50 | 0.55 | 0.596 |

Figure 6 shows some predicted VFFs in comparison to those manually digitized in the Val, KY1, KY2, and VA study areas. The figure suggests that VFFs were generally detected with few FPs. Although the

overall shapes of the VFFs are similar in the automated and reference (i.e., manually digitized) datasets, the boundaries do not overlay exactly due to their fuzzy nature, as previously discussed.

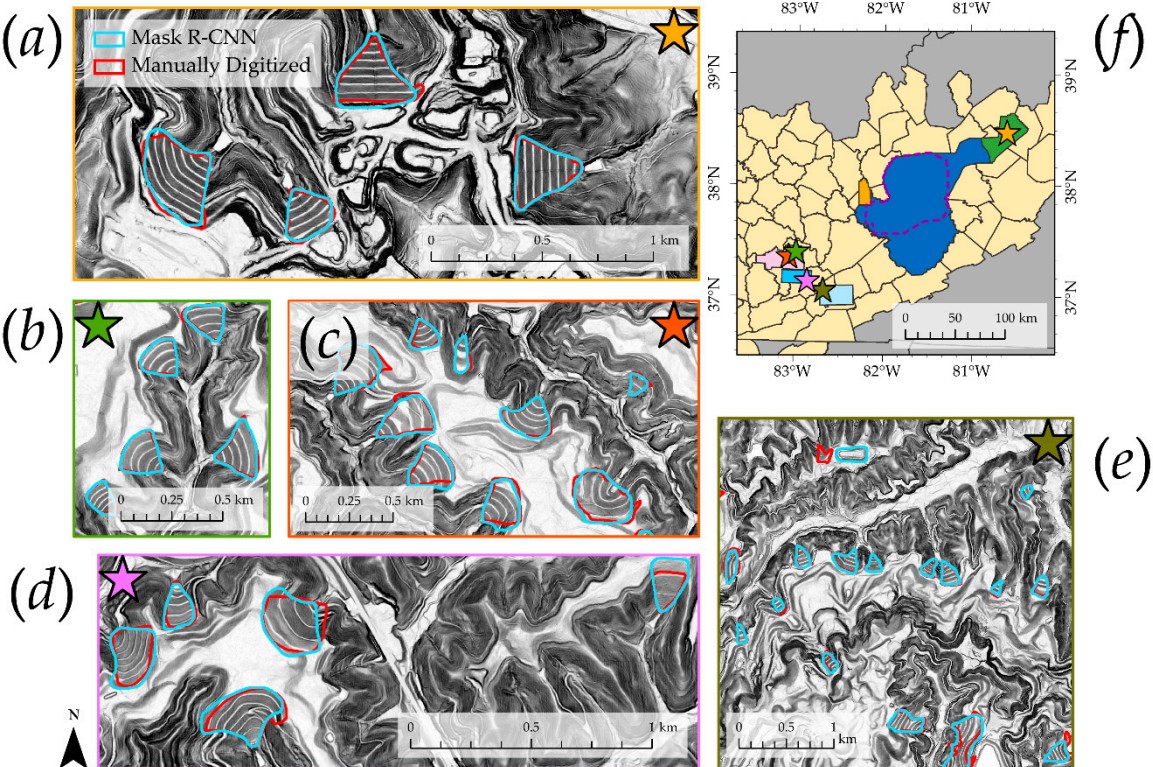

**Figure 6.** Example of VFF predictions and manually digitized features across multiple study areas. Stars indicate the location of (*a*) through (*e*) in the study areas (*f*).

Figure 7 provides some examples of common FP issues. Some reclaimed mine sites, coal and overburden piles in mine sites, and artificially re-contoured landscapes associated with residential or transportation development, were occasionally misclassified as VFFs (Figure 7*b–d*). Slopes with timber-harvest roads (Figure 7*a*), which result in a pattern similar to terracing, especially within valley-head areas, were sometimes falsely mapped as VFFs. Figure 7*e* shows a misclassified slope that is characterized by deep channeling, which has potentially been confused as the drainage ditches installed on the VFFs. Mapped VFFs that were most often missed (FNs), were those that covered a smaller area and/or lacked the characteristic terracing pattern or drainage ditches. In contrast, VFFs that were larger and had well defined terracing where seldom missed.

*3.2. Validation*

Table 4 shows the validation based on the manual inspection of TP, FP, and FN VFF counts in all areas that were predicted. For predictions made using the LiDAR-derived data, precision, recall, and F1 score values were all higher than 0.73. The highest precision was obtained for the KY1 area while the highest recall was obtained for the Val area. In general, we documented similar precision, recall, and F1 scores for the LiDAR-derived data, suggesting that the model generalizes well to other geographic regions, collected using different LiDAR sensors. Further, a higher precision was obtained for the KY1 and KY2 test sites than the Val site, which was mapped using the same LiDAR sensor as the data used to develop the model. The VA test site had the lowest precision, recall, and F1 scores of all the LiDAR-derived extents. This could potentially be a result of the higher average post spacing in comparison to the other collections, resulting in a disparate representation of the landscape. Another confounding factor is the characteristics of the VFFs in each area. For example, a visual

inspection of the KY1 extent suggests a large number of VFFs that are large and have well defined terracing, which may contribute to comparatively high assessment metrics for this extent. The All LiDAR column represents the pooled results for all LiDAR-derived datasets. Collectively, a precision of 0.878, a recall of 0.858, and a F1 score of 0.868 was obtained when using LiDAR-derived data. For VFFs larger than 1 ha, recall increased for all the study sites, which indicates that the larger fills were generally easier to detect.

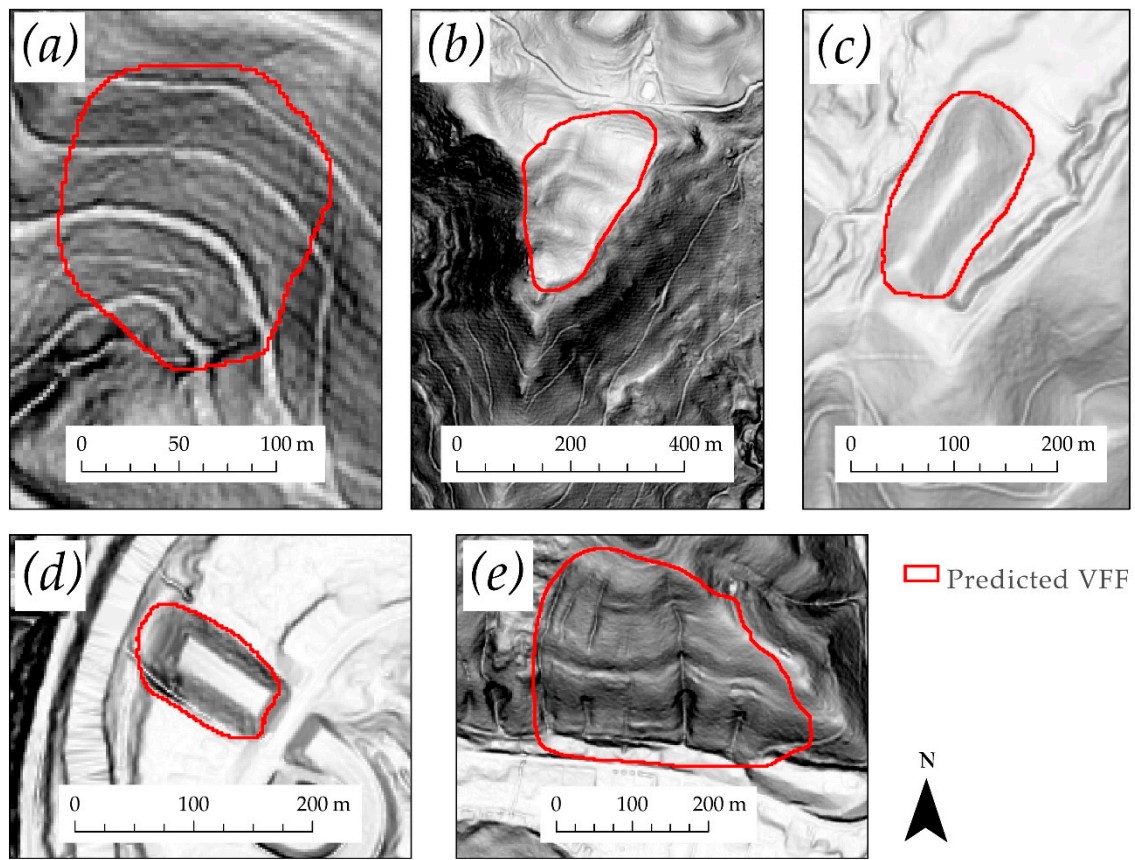

**Figure 7.** (*a*) through (*e*) Example of false positives.

**Table 4.** Validation based on manual comparison of numbers of VFFs.

| Measure | Val | KY1 | KY2 | VA | All LiDAR | SAMB1 | SAMB2 |
|---|---|---|---|---|---|---|---|
| | \multicolumn{7}{c}{**Study Area**} | | | | | | |
| No. Mapped VFFs | 182 | 540 | 149 | 143 | 1014 | 581 | 108 |
| No. Mask R-CNN VFFs | 200 | 546 | 143 | 149 | 1038 | 1735 | 321 |
| TP | 170 | 495 | 129 | 117 | 911 | 346 | 39 |
| FP | 30 | 51 | 14 | 32 | 127 | 1389 | 282 |
| FN | 13 | 59 | 37 | 42 | 151 | 239 | 69 |
| Precision | 0.850 | 0.907 | 0.902 | 0.785 | 0.878 | 0.199 | 0.121 |
| Recall | 0.929 | 0.894 | 0.777 | 0.736 | 0.858 | 0.591 | 0.361 |
| F1-Score | 0.888 | 0.9 | 0.835 | 0.76 | 0.868 | 0.278 | 0.181 |
| No. Mask R-CNN VFFs (>1 ha) | 123 | 463 | 129 | 111 | 826 | 527 | 77 |
| TP (>1 ha) | 118 | 418 | 105 | 89 | 730 | 315 | 35 |
| FN (>1 ha) | 5 | 48 | 2 | 25 | 80 | 217 | 42 |
| Recall (>1 ha) | 0.959 | 0.897 | 0.809 | 0.781 | 0.900 | 0.592 | 0.455 |

Precision, recall, and F1 score were generally low when the model trained on the LiDAR-derived data was used to predict to the photogrammetrically-derived slopeshades in the SAMB1 and SAMB2 extents. All values are lower than 0.6 due in part to the large number of FPs. A visual inspection of

the classifications suggests that larger VFFs with well-defined terracing were generally mapped well; however, many VFs were missed, especially smaller features or those without well-defined terracing. Since the photogrammetric methods due not allow for canopy penetration, VFFs that were heavily vegetated with woody vegetation and shrubs generally did not show well defined terracing in the slopeshades even if the pattern was present. Figure 8 provides examples of manually digitized VFFs as represented in the photogrammetric data. Note that the characteristic terracing pattern is not evident for all features. It would be interesting to assess models trained form the photogrammetric data to assess the ability to map historic conditions from such data; however, that is outside the scope of this study, as our goal here is to assess the transferability of the model trained on LiDAR-derived data to disparate datasets.

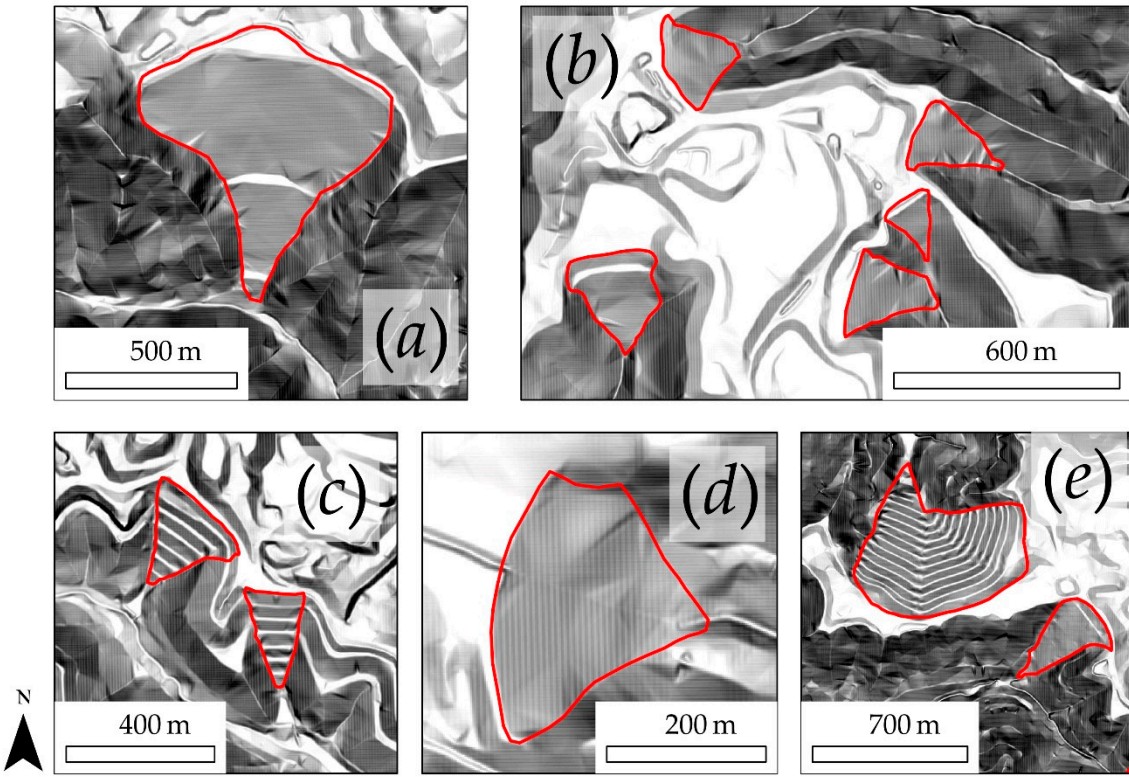

**Figure 8.** (*a*) Through (*e*) Examples of digitized valley fills as represented in the photogrammetric data.

In summary, these results indicate that the model performed well using LiDAR-derived data; however, photogrammetric data resulted in many FPs and generally poor performance based upon precision and recall. The model was not able to adequately generalize to these very different datasets. It should be noted that it is difficult to note whether or not the poor performance is a result of the inability of the model to generalize to the disparate data or whether it is because the VFFs are not well represented in the photogrammetric datasets.

Validation using area-based user's and producer's accuracy (Table 5) generally yielded lower assessment values than the count-based evaluation (Table 4). For example, all producer's and user's accuracies for the predictions using LiDAR-derived data were lower than the associated count-based recall and precision values, respectively, except for KY2. The photogrammetric data resulted in user's and producer's values between 0.043 and 0.388.

Using the fuzzy, center-weighted method, user's and producer's accuracies increased for all LiDAR-derived results in comparison to the area-based method. This highlights that the center portions of the VFFs were generally mapped well, and the boundaries, which are inherently indeterminate,

had lower agreement. Notably, the aggregate All LiDAR measures indicate a fuzzy producer's accuracy of 0.860 and a fuzzy user's accuracy of 0.903.

**Table 5.** Area- and fuzzy area-based error evaluation.

| Measure | Study Area | | | | | | |
| --- | --- | --- | --- | --- | --- | --- | --- |
| | Val | KY1 | KY2 | VA | All LiDAR | SAMB1 | SAMB2 |
| Producer's Accuracy (Area) | 0.787 | 0.797 | 0.831 | 0.735 | 0.793 | 0.129 | 0.263 |
| User's Accuracy (Area) | 0.841 | 0.741 | 0.78 | 0.603 | 0.744 | 0.388 | 0.043 |
| Fuzzy Producer's Accuracy (Center-Weighted) | 0.851 | 0.866 | 0.899 | 0.802 | 0.860 | 0.137 | 0.046 |
| Fuzzy User's Accuracy (Center -Weighted) | 0.909 | 0.944 | 0.964 | 0.689 | 0.903 | 0.169 | 0.053 |

## 4. Discussion

### 4.1. Study Findings

Our results, as highlighted by our demonstration of mapping VFFs, suggest that Mask R-CNN can be used to extract terrain features from digital elevation data if the features have a unique topographic signature. Count-based precision, recall, and F1 scores were all above 0.73 across all validation datasets when predicting to LiDAR data. When assessed based on area, producer's and user's accuracies were generally lower than the associated recall and precision values. This highlights the difficulty in mapping and assessing boundaries that are poorly defined or gradational in nature. When center-weighting the areas to reduce the influence of boundaries, we saw increases in user's and producer's accuracies, which suggests that the features were generally successfully mapped, though the boundaries may be disparate. It is notable that this model generally had similar performance when applied to new geographic regions, as long as LiDAR data were used, even if the sensors and acquisition parameters differed. This provides strong evidence that the approach is robust and can be applied to regional mapping. However, the model performed poorly when applied to disparate, photogrammetrically-derived data, suggesting that generalization is limited to comparable data. However, this may partially result from the poor representation of the VFFs in the photogrammetric data, so it is difficult to differentiate the impact of the model and the input data quality. Overall, this case study shows promise in applying instance segmentation to digital terrain data, suggesting that CNN-based deep learning has potential for mapping other topographic classes, for example geomorphological or soils mapping.

Since no prior studies have attempted to map VFFs using automated methods, we are not able to relate our findings to any prior studies that have explored this specific task; however, our findings do reinforce those of Tier et al. [6] and Behrens et al. [48], which note the value of CNNs for extracting features from digital terrain data. More broadly, this study supports prior findings that CNNs in general and Mask R-CNN specifically are of great value for mapping features with a unique spatial, contextual, or textural signature and that may not be spectrally separable from other classes or features [51,52,84,85].

### 4.2. Limitations and Recommendations

There were some notable limitations in this study. Although we analyzed multiple study sites and datasets, this case study is specific to a single geomorphic feature. Therefore, it would be useful to assess the application of instance segmentation techniques to map and differentiate additional anthropogenic and natural terrain features. For example, these methods could be applied to mapping glacial, fluvial, or eolian landforms. Additionally, there is a need to assess the mapping of landscape change using multitemporal digital terrain data. This could not be pursued in this study due to

the lack of pre-mining LiDAR data and our finding that photogrammetric data were not useful for mapping VFFs.

Given the large training dataset requirements of deep learning, there is a need to develop databases of training samples specific to terrain data and features, similar to those that already exist for photographs, such as MS COCO [72] and ImageNet [86]. The remote sensing community should consider investing in the development of large terrain- and image-based datasets to improve our ability to apply deep learning to our data and perform more robust experiments. Trier et al. [6] made a similar argument within the archeological research community. Transfer learning from models trained on terrain data may prove more accurate than the application of weights learned from RGB photographs.

In this project, we specifically relied on slopeshades as a representation of the terrain surface. There is a need to explore additional representations or combinations of representations as input to DL techniques. With LiDAR, it is possible to obtain measures of the height of above ground features, such as trees and buildings, by generating a normalized digital surface model (nDSM). It is also possible to measure the intensity of the returned laser pulse [10]. There is a need to explore these additional measures for mapping terrain features with DL. As noted by Stereńczak et al. [87], the interpolation method used to generate a DTM from LiDAR point cloud data can have an impact on the resulting representation of the terrain surface, so there is also a need to assess how well models perform and transfer to DTMs generated using different interpolation methods.

Due to lengthy computational time and the large number of parameters that can be manipulated, it was not possible to fully optimize the Mask R-CNN method for this task. Training the model for 24 epochs using a single GPU required 12 h; as a result, we could not extensively experiment with the impact of parameter settings as would be possible with traditional, shallow machine learning methods using grid searches combined with *k*-fold cross validation or bootstrapping. Instead, we had to rely on a limited series of experiments using a small number of model parameters. This issue will need to be addressed in order to support rigorous comparative studies of different algorithms and/or feature spaces. Terrain feature extraction could be performed with semantic segmentation methods, such as SegNet and UNet, so there is a need to compare different CNN methods for mapping terrain features.

It is common for landscape features to have boundaries that are gradational or inherently indeterminate, which adds to the complexity of assessing the quality of the prediction. A review of the remote sensing, computer vision, DL, and machine learning literature suggests a lack of research on this topic. Our method is an extension of an approach proposed by Brandtberg et al. [83] and appears to have potential for widespread use in applications such as mapping wetlands and soils, as well as tree delineation from high resolution images. However, it would be useful to explore this approach more thoroughly, as additional refinements may improve it. For example, measures of distance other than a linear approach may be useful.

## 5. Conclusions

This exploration of mapping VFFs from digital terrain data suggests that the Mask R-CNN DL instance segmentation method can be applied to map geomorphic and landscape features using LiDAR-derived data. Further, our results suggest that models trained in one area can transfer well to other areas if similar data are available, in this case LiDAR, providing strong evidence of the robustness of the approach. However, the model performed poorly when applied to disparate, photogrammetrically-derive data.

Here we focused on features that have distinctive topographic and geomorphic characteristics, and we suggest that there is a need for further experimentation relating to mapping additional terrain features of variable complexities. Future studies should explore the mapping of additional anthropogenic, fluvial, glacial, and eolian terrain features and landforms. There is also a need to further explore optimization methods for deep learning to foster more rigorous comparisons and develop standardized techniques to assess gradational or uncertain boundaries. As CNN-based semantic and

instance segmentation methods mature, there is a need to further explore these techniques for mapping and extracting features from geospatial and remotely sensed data.

The recent developments in CNNs, semantic segmentation, and instance segmentation are providing new opportunities to extract and map digital terrain features. Our hope is that this case study will encourage additional research and data development relating to automated terrain mapping using LiDAR and DL. Further, with the increasing availability of LiDAR data, such methods will likely prove to be of great importance for studying our anthropogenic and natural landscapes and monitoring landscape change.

**Author Contributions:** Conceptualization, A.E.M.; methodology, A.E.M., and P.P.; validation, A.E.M. and J.D.P.; formal analysis, A.E.M. and J.D.P.; writing—original draft preparation, A.E.M.; writing—review and editing, A.E.M., P.P., and J.D.P. All authors have read and agreed to the published version of the manuscript.

**Funding:** No funding obtained for this project.

**Acknowledgments:** We would like to acknowledge the West Virginia GIS Technical Center, West Virginia View, and the United States Geologic Survey (USGS) who provided LiDAR data for this study. We would also like to thank five anonymous reviewers whose suggestions and comments strengthened the work.

**Conflicts of Interest:** The authors declare no conflict of interest.

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
