# Peer review of "Mapping the Topographic Features of Mining-Related Valley Fills Using Mask R-CNN Deep Learning and Digital Elevation Data"

_remotesensing, doi:10.3390/rs12030547_

Round 1
Reviewer 1 Report
Summary: The manuscript evaluated the performance of a DL model-Mask R-CNN in automatically mapping valley fill faces (VFFs) from digital terrain data in the Appalachian region of the eastern United States. The results show that (1) the model is an excellent model in mapping VFFs; (2) the model works better with LiDAR-derived slopeshades compared to photogrammetrically-derived slopeshades. A major concern on the structure of the manuscript must be addressed before considering for publication. Therefore, I suggest a major revision.
Major concern:
There might be a better way to introduce/mention the use of photogrammetrically-derived slopshades. I thought the paper only tested the model with LiDAR-based on the title of this manuscript. In fact, I think this comparison experiment plays an important role in this article. Photogrammetrically-derived digital terrain dataset encompasses a longer time frame of data so that there might be an interest to map the historic footprint of VFFs from photogrammetrically-derived digital terrain datasets. However, this study proves that Mask R-CNN does not perform well with it.
Other comments:
[1] line 16: I suggest that the authors use Mask R-CNN instead of mask R-CNN. That's how He et al. named their model in the 2017 paper. This applies to the rest of the paper.
[2] line 73: Similar to my major concern, I did not see anything related to the photogrammetrically-derived digital terrain dataset until the last sentence of section 1. Introduction. I do not have an exact suggestion for the authors but I would suggest the authors find a better way to make the dataset less "surprising".
[3] line 91: OBIA is a more common name to be used to represent "object-based image analysis". [19] used OBIA as well.
[4] line 153: I would suggest authors use "125 million building footprints".
[5] figure 1: The order of the figures is sort of odd.
[6] line 254: Related to the major concern again, it seems like the training data include photogrammetrically-derived digital terrain dataset. But the use of photogrammetrically-derived digital terrain dataset in mapping VFFs was barely mentioned in the literature review compared to LiDAR.
[7] line 334: I am not sure if this is a common way to call "tile" or "subset" as “chip”. The authors used "tiles" in the line 343 as well. I suggest the authors keep the use of terms as consistent as possible.
[8] line 471: Did the authors set IoU as >0.5 or >0.55? Please clarify why there could be a range of IoU thresholds (fuzzy rules)?
[9] lines 479-485: Did the authors use any negative training samples to prevent the classifier from misclassifying those valley-head areas etc? That might improve the Mask R-CNN performance.
[10] figure 6(f): A scale bar is missing.
[11] figure 6: Again, the order of the figures is strange.
[12] lines 514-515: This issue again. I feel like the authors really wanted to point out that LiDAR is better. But the results from this experiment should be better prepared and presented since it was part of this study.
Reviewer 2 Report
The manuscript present a method to identify Vallay Fill Faces (VFF) for mining application with MasK R-CNN. Although it is well prepared and results are somewhat interesting, unfortunately it is difficult for the reviewer to find significant or novel scientific materials in the manuscript. The reviewer suggests to directly investigate or modify the original CNN, e.g. adding the new layer to the original to deal with the authors' datasets, to be worth published in the journal.
Reviewer 3 Report
Reviewed article „Mapping the Topographic Features of Mining-Related Valley Fills using Mask R-CNN Deep Learning and Digital Elevation Data„ raises extremely important issues related to the reconstruction and modeling of topographic features associated with large-scale human surface transformation. This is an important issue due to the significant transformation of the earth's surface by man, on the one hand, and also the increasing and common access to high resolution elevation data (LiDAR type). In this study, the authors decided the use of mask R-CNN for mapping valley fill faces (VFFs) resulting from mountaintop removal (MTR) surface coal mining. The authors set themselves a goal two objectives: 1) assess the mask R-CNN DL algorithm for mapping VFFs using LiDAR-derived digital elevation data, and 2) investigate model performance and generalization by applying the model to LiDAR-derived data in new geographic regions and acquired with differing LiDAR sensors and acquisition parameters, as well as a photogrammetrically-derived digital terrain dataset.
The authors decided to use the deep learning (DL) method in their research, because they showed its high efficiency in the literature review. In addition, it turned out that it is lack of research associated with mapping terrain features from digital terrain data using DL methods, and no published studies that apply this algorithm to raster-based digital terrain data for mapping geomorphic features.
The authors thoroughly presented the introduction - providing evidence of knowledge of the latest literature on the subject. In addition, when it comes to the methods used - they thoroughly prepared the entire calculation procedure, because: 1) the model was trained over a single large area, 2) then tested over an adjacent smaller area, 3) next used to make predictions in a new area collected with the same LiDAR sensor. Moreover three additional extents in different states mapped during different LiDAR collections, and two extents of photogrammetric data within West Virginia.
The authors described the research process in detail, and described the learning model. The method's errors and limitations were also tested and presented. This is important for the future application of the method to other areas and for other data sets. They noted that even for data from the same LiDAR sensor, the recognition efficiency of the distinguished forms was different. This, of course, results from the topographic diversity and, as the authors noted, from the size of the studied forms (large forms, over 1 ha were better detected). In total, quite high precision of of 0.878, a recall of 0.858, and a F1 score of 0.868 was obtained when using LiDAR-derived data.
An important working conclusion of the authors is the statement that CNN-based deep learning has potential for mapping other topographic classes, for example geomorphological or soils mapping. I also agree with the authors that with the increasing availability of LiDAR data, such methods will likely prove to be of great importance for studying our anthropogenic and natural landscapes and monitoring landscape change.
To sum up - the article is written correctly, the layout of the content does not raise any objections. Stages of work are presented in a clear and transparent way. The results have been well commented in the "Discussion" section, so one can orient on the strengths and weaknesses of the used method. After careful reading of the article, I have no major comments and I think that the text in its current form is suitable for publication.
Reviewer 4 Report
I would like to congratulate all the authors very well written article. It's rare situation to review so interesting article so well written. I have nothing to add except one thing maybe. Although plenty of different ALS data sets were used, Authors did not generated DTM by themselves. This is not a complaint but it is worth to keep in mind that different algorithms used to generate DTM may give slightly different accuracy (described by Sterenczak et all., 2016) and in consequence improve your method.
Reviewer 5 Report
Dear authors,
I have revised the paper and have several comments for improving the quality of the manuscript as below:
In Abstract, please give some quantitative results for it. Introduction is too long, you should shorten it and make a better story for it. Moreover, I do not prefer to divide the Introduction part into several sub-sections, please combine them. I would suggest authors to compare CNN with several other benchmark machine learning models like ANN, SVM or hyrbid models. Remove results word in "Validation results" of section 3.2 and modify the title of this section. Discussion is not enough, you should extend the discussion by comparing your findings with previous published papers. What is future works should be given in the Conclusion part.Best of luck!
Round 2
Reviewer 1 Report
The authors have addressed my concerns. I can recommend for a publication.
It would be great if the authors can come up with a better solution to fix the order issues of the subfigure.